# Adoptive Cellular Therapy for Multiple Myeloma Using CAR- and TCR-Transgenic T Cells: Response and Resistance

**DOI:** 10.3390/cells11030410

**Published:** 2022-01-25

**Authors:** Franziska Füchsl, Angela M. Krackhardt

**Affiliations:** 1School of Medicine, Klinik und Poliklinik für Innere Medizin III, Klinikum rechts der Isar, Technische Universität München, Ismaningerstraße 22, 81675 Munich, Germany; franziska.fuechsl@tum.de; 2German Cancer Consortium (DKTK), Partner-Site Munich, and German Cancer Research Center (DKFZ), 69120 Heidelberg, Germany; 3Center for Translational Cancer Research (TranslaTUM), School of Medicine, Technical University of Munich, Einsteinstraße 25, 81675 Munich, Germany

**Keywords:** multiple myeloma, adoptive cellular therapy, CAR-T cells, TCR-T cells, T cell engineering

## Abstract

Despite the substantial improvement of therapeutic approaches, multiple myeloma (MM) remains mostly incurable. However, immunotherapeutic and especially T cell-based approaches pioneered the therapeutic landscape for relapsed and refractory disease recently. Targeting B-cell maturation antigen (BCMA) on myeloma cells has been demonstrated to be highly effective not only by antibody-derived constructs but also by adoptive cellular therapies. Chimeric antigen receptor (CAR)-transgenic T cells lead to deep, albeit mostly not durable responses with manageable side-effects in intensively pretreated patients. The spectrum of adoptive T cell-transfer covers synthetic CARs with diverse specificities as well as currently less well-established T cell receptor (TCR)-based personalized strategies. In this review, we want to focus on treatment characteristics including efficacy and safety of CAR- and TCR-transgenic T cells in MM as well as the future potential these novel therapies may have. ACT with transgenic T cells has only entered clinical trials and various engineering strategies for optimization of T cell responses are necessary to overcome therapy resistance mechanisms. We want to outline the current success in engineering CAR- and TCR-T cells, but also discuss challenges including resistance mechanisms of MM for evading T cell therapy and point out possible novel strategies.

## 1. Cellular Therapy in Multiple Myeloma

Multiple myeloma (MM) remains an incurable B cell malignancy in many patients although the advancement of novel therapeutic approaches is constantly improving the outcome of this disease. However, most patients are relapsing and survival in these patients is often short, especially for triple refractory patients progressing after receiving multiple lines of proteasome inhibitors (PI), immunomodulatory drugs (IMiDs) and anti-CD38 treatment [1,2]. Thus, novel therapeutic approaches are urgently needed. Cellular therapy represents a treatment strategy, which has shown great success in the treatment of B cell leukemias and lymphoma especially by targeting CD19 using chimeric antigen receptor (CAR) T cells. Within multiple clinical trials, high and durable responses were achieved in patients suffering from acute lymphocytic leukemia or B cell non-Hodgkin lymphoma after infusion of T cells engineered to express this synthetic receptor [3,4,5]. Attempting to reach similar responses in MM patients, B cell maturation antigen (BCMA) targeting CAR-constructs has been developed with impressive results. Idecabtagene vicleucel (ide-cel, also called bb2121) [6,7] was recently approved by the FDA and EMA for clinical application in patients with relapsed and refractory MM. These developments pave the way for broader application of T cell-based adoptive cellular therapies (ACT) in MM which are not limited to CARs. As artificial chimeric fusion receptors CARs are empirically designed to mimic signaling downstream antigen-specific T cell receptor (TCR) stimulation. However, the diversity and adaptive potential of a T cell response are likely not reflected by these constructs. More physiological T cell signaling may be achieved by equipping patient T cells with tumor reactive TCRs (Figure 1A–D).

ACT is not a completely new concept in MM: non-genetically modified cell products on the one hand comprise allogeneic stem cell transplantation (SCT) or autologous lymphocyte infusions, including particularly administration of marrow infiltrating lymphocytes (MIL). Both approaches are exploiting endogenous myeloma-reactive T cells—as well as the less abundant natural killer (NK) cells—to enable tumor recognition [8,9]. The alternative, on the other hand, covers all forms of genetically modified cell products. Those mainly comprise transgenic T cells engineered to either express a natural TCR (Figure 1A) targeting tumor associated antigens (TAA) or neoantigens or a CAR (Figure 1B–D) targeting a specific antigen on the tumor cell surface in its native conformation (Figure 1E). Both receptors aim for potent T cell activation with subsequently efficient tumor cell killing as well as the initiation of stable, long-term immune memory for tumor control [10,11,12]. Alongside approved antibody- [13,14,15] or still experimental vaccine-based [16,17] treatment options, transgenic T cell-based ACT harbors great hope for long term tumor surveillance and complete, stable disease control in MM patients [18,19].

## 2. CAR-T Cell Therapy in MM and the Evolving Generations of CARs

### 2.1. General Aspects of CAR Constructs

CARs are designed to mimic the signal pathway downstream the native TCR based on our understandings of T cell signaling. Therefore, a CAR traditionally consists of an antigen recognition domain (a single-chain Fv (scFv), containing the variable domains of light and heavy chain of an antibody), a spacer and transmembrane region as well as a signaling domain. The latter comprises for all, so far, clinically approved CARs the CD3ζ chain as well as the CD28 (28ζ) or 41BB (41BBζ) (also known as CD137) intracellular costimulatory domain [20].

While the complete native CD3 complex is built of six subunits with ten total immunoreceptor tyrosine-based activation motifs (ITAMs) involved in signal transduction [21], it has been shown that singular subunits, such as the CD3ζ chain, alone induce signaling events identical to those downstream of TCR ligation in T cell hybridomas in vitro [22,23]. The first attempts towards artificial fusion receptors built from antibody recognition domains and TCR-signaling chains were conducted three decades ago with CARs containing the CD3ζ-chain [24] or the CD3γ-chain alone [25,26]. These first-generation CARs (see Figure 1B) were sufficient to induce cell lysis, but not sustained tumor control due to insufficient signal strength for the activation of resting T cells [27].

Beyond this core element of imitating TCR signaling, the addition of one [3,28,29,30] and then another [31,32,33] costimulatory domain, respectively CD28 or 4-1BB, paved the way up to the third generation of CAR-constructs substantially improving the transgenic T cells’ efficacy (see Figure 1B,C). The choice of the the costimulatory domain, mostly investigated in CD19 CARs, thereby influences antigen recognition sensitivity, strength of T cell activation, longevity and clinical applicability: 28ζ stimulatory domains seem to be more sensitive in antigen recognition [34], produce larger amounts of cytokines [32,35], rely mostly on oxidative glycolysis, cause more rapid expansion and induce a rather effector-like phenotype [36]. 41BBζ CARs, on the contrary, reach prolonged T cell persistence in vitro, predominantly metabolize fatty acids and maintain a more central memory-like phenotype—with some of these effects appearing to be antigen-independent [31,32,36,37].

Overall response rates for CARs directed against CD19 in adults with relapsed or refractory B cell lymphoma are, so far, in a comparable range despite different costimulatory domains. Yet, further investigations are currently ongoing for different entities and indications. Two second generation constructs are approved for clinical use in the EU, so far: axicabtagene ciloleucel engineered to express a CD28 co-stimulatory domain and tisagenleucel with 41BB co-stimulation [3,30]. Longer T cell persistence for 41BB-CARs is suggested by clinical follow-up of infused CAR products. Persistence for roughly 30 days for CD28ζ-CARs in acute lymphocytic leukemia [38] and for up to 4 years for 41BBζ-CAR constructs in chronic lymphocytic leukemia [39] were reported. However, a clear comparison between patient cohorts is not possible and entity-associated factors may play a role. An additional aspect to compare is the rate of severe side effects, such as cytokine release syndrome (CRS) and immune effector cell associated neurotoxicity (ICAN). Both remained lower for 41BBζ CARs in preclinical studies. Meta-analyses of 40 clinical studies, however, could not report these differences in toxicity profile [33,40,41].

Attempting further enhancement of CAR-T cell functionality and longevity, a third generation of CARs (see Figure 1C) is engineered nowadays by combining two co-stimulatory elements in one intracellular domain—mostly the above mentioned CD28 and CD137 signaling site [31,32,42]. Some advantages in a proliferative capacity, increased activation of intracellular signaling pathways over their predecessors in vitro, and elevated anti-tumor control in pre-clinical in vivo models [43,44] could be reported. However, few clinical trials have investigated these constructs in the patient so far, and clear superiority to second generation CARs has not been published yet [31,45,46].

Further engineering, nevertheless, has already yielded a fourth generation of CAR constructs. The addition of inducible IL-12-release at the tumor site by so called “armored” CARs, also known as TRUCKS (“T cell redirected for antigen-unrestricted cytokine-initiated killing”), shall further enhance efficacy in solid and at least partially antigen-negative tumors by attracting other immune cells, especially from the innate immune system, to the tumor site [47,48,49]. Additional adaptations under current investigation include further co-stimulatory domains [50] or integrated suicide switches for better control of adverse events [51].

So far, most of these investigations employed CD19-CAR models to enhance construct efficacy. Whether the choice of optimal signal domain composition or the addition of internal autocrine cytokine release and suicide switches depend on the tumor entity, qualitatively or quantitatively on the target antigen or on other factors, requires further investigation for other malignancies and constructs—such as BCMA CARs in MM [32]. However, while these artificial receptors are mainly considered a breakthrough in tumor therapy, their potential for application in other disease settings, exemplarily infectious [52] or autoimmune [53,54], should also be mentioned.

### 2.2. Potential Target Structures for CAR-Therapy in MM

After the success of CD19 CARs for the treatment of B cell lymphoma, MM, another immune cell derived cancer entity, displays potential for successful CAR therapy. Especially patients relapsed after at least three to four therapy lines, non-transplant eligible as well as high risk patients might benefit from CAR therapy as an alternative to already established therapy regimens. One substantial aspect of successful CAR-T cell administration is the identification of suitable target structures on the surface of malignant cells. For myeloma, several surface markers, all preferentially expressed on mature B cells, qualify as potential candidates (see Table 1 and Figure 1E) which have been reviewed elsewhere in more detail [55]. The target amongst them, which reached most attention and which shall also be in the focus of this review, is BCMA. In the meantime, CD19 should not be neglected as a potentially attractive target for MM therapy [56] and recently, also SLAMF7-specific CAR constructs are increasingly tested in the clinic [55,57].

Important, when choosing a target for ACT, is its specificity for the tumor tissue. Cell-lineage markers such as CD19, CD20 as well as BCMA—the currently most common target antigens in CAR-therapies—are not ideal in terms of specificity. Healthy B cells and their precursors, if not already cleared by high dose chemotherapy before ACT, are also targeted by reactive CARs. B cell aplasia, the resulting on-target off-tumor toxicity, however, is manageable with antibiotics and/or infusion of immunoglobulins but can though impact morbidity if persisting over a longer period [58,59]. Compared to these relatively mild consequences, a lack of specificity can, however, be fatal for other CAR constructs. After the application of an ERBB2-recognizing CAR construct in a colon cancer patient, severe respiratory distress immediately after CAR-T cell-infusion with eventually lethal CRS, despite previously high in vitro specificity, was reported. Dramatic pulmonary infiltrates suggest the recognition of low levels of ERBB2 on lung epithelium as the cause [60]. On-target off-tumor effects, therefore, must be assessed for their toxicity and can be tolerated only, if outweighed by the clinical benefit. Though specificity of BCMA suffices for effective use of CAR-T cells, increasing specificity of CAR constructs in MM ultimately aims at more specific targets for malignant cells only. Exemplarily, G-protein–coupled receptor class C group 5 member D (GPRC5D) is expressed BCMA-independently on CD138+ MM cells, but only minimally in healthy tissue, except for hair follicles [61]. First-in-human studies showed a manageable safety profile as well as promising efficacy in highly pretreated patients [62].

### 2.3. Targeting BCMA in MM CAR-Treatment

Like the above-mentioned Ide-cel, most CAR-approaches for MM to date focus on BCMA (also referred to as TNFRSF17 or CD269), a member of the tumor necrosis factor receptor (TNFR) superfamily. BCMA is predominantly expressed on maturated B cells and therefore plasma cells, and plays a role in B cell development, while its presence is not obligatory for their maturation [63,64,65,66]. Its two ligands are B cell activating factor (BAFF, also known as BLyS)—which also binds to the transmembrane activator and CAML interactor (TACI) and B cell activating factor receptor (BAFF-R)—and a proliferation inducing ligand (APRIL). Both are known to influence MM cell growth [67]. The consistent expression on the MM cell surface—albeit at various levels in different patients—renders BCMA a suitable target antigen for CAR therapy [67,68,69].

Several clinical trials (see selection in Table 2) were or are investigating the efficacy and safety of BCMA-targeting T cells. The first-in-human study of a BCMA-CAR was performed with a construct incorporating a mouse scFv (11D5-3), CD8α-hinge/-transmembrane domain, a CD28 co-stimulatory domain and a CD3ζ-chain [70]. The median event-free survival duration of this study was 31 weeks, however, all patients developed disease progression [70].

Ide-cel, the recently approved CAR product, integrates the same murine scFv for antigen recognition but is designed with a 4-1BBζ intracellular stimulatory element. In phase I clinical trials, the median progression free survival (PFS) ranked at 11.8 months with an overall response rate (ORR) of 85% [7,71]. In the following pivotal phase II KarMMa study (NCT03361748), 128 relapsed and refractory patients after three to sixteen (median of six) preceding therapy lines, received ide-cel infusion (of 140 total enrolled patients) [6]. The study supports substantial antitumor activity of ide-cel in heavily pre-treated patients with an ORR of 73%, complete response (CR) of 33% and a PFS of 8.8 months over all applied doses (slightly better outcomes occurred for the highest dose chosen). In 36% of all patients, CAR-T cells were detectable in the blood after 12 months, which, however, did not protect from relapse. At the time of relapse, most patients still had detectable levels of BCMA on their tumors, which renders antigen loss an unlikely sole cause for tumor progression. Adverse events, especially transient hematologic toxic events, were detected in all and those of grade 3 or 4 in almost (127 of 128 patients) all patients. CRS occurred in 84% of cases (grade 3 to 4 in 5% only), and neurotoxic effects were seen in 18% [6].

A similarly heavily pretreated patient group (*n* = 97) received CAR-T cell infusion with Ciltacabtagene autoleucel (Cilta-cel, also known as LCAR-B38M and JNJ-68284528), a 41BBζ-construct with two alpaca-derived single domain BCMA-recognizing antibodies, during the combined phase Ib/II clinical trial CARTITUDE-1 (NCT03548207). An ORR of 97% and CR of 67% were reported at a manageable safety profile with numerous hematologic adverse events, but few cases of severe (grade 3–4) CRS (4%) or neurotoxicity (9%). PFS was not reached at clinical cutoff after a median follow-up time of 12.4 months [72,73]. Drawing a comparison between the patients enrolled in this single-arm study and an external, comparable study cohort representative for real-world data for current MM standards of care, a significant improvement in ORR, PFS and overall survival (OS) could recently be reported for CAR T cell treatment (LocoMMotion, NCT04035226) [74]. For further assessment of minimal residual disease rates upon Cilta-cel infusion, patients for CARTITUDE-2 are currently recruited (NCT04133636). Moreover, the previous success of Cilta-cel led to an extended investigation in phase III clinical studies: CARTITUDE-4 investigates the effects of Cilta-cel when administered to relapsed and lenalidomide-refractory patients (NCT04181827), while the meanwhile launched CARTITUDE-5 study focusses on the combination treatment of Bortezomib, Lenalidomide and Dexamethasone (VrD) with Cilta-cel in newly diagnosed MM patients (NCT04923893).

Despite all success, it becomes evident that the plateau phase of tumor progression as seen after CD19-CAR administration for B cell lymphoma [75,76] has not been achieved in a similar fashion with BCMA-CARs in MM so far. Further adjustment and engineering of the CAR-constructs themselves, the dose or administration scheme therefore might be considered for improving CAR-T cell-performance, tumor killing and control.

**Table 2 cells-11-00410-t002:** Selection of important clinical trials for BCMA-CAR transgenic T cells (information from clinicaltrials.gov).

Trial Number/Name	Sponsor	CAR Construct	Phase	n ^1^	Origin of scFv	Co-Stimulatory Domain	Dose ^2^	Conditioning Therapy ^3^	ORR ^4^	CRS ^4^ (All/ gr 3–4)	ICAN ^4^ (All/ gr 3–4)	Further Modifications/ Comments	References
NCT02215967	National Cancer Institute (NCI)		I	30	murine	CD28	0.3–9.0 × 10^6^	CP/ Flu	81%	94%	N.A.		[70]
NCT03070327	Memorial Sloan Kettering Cancer Center	MCARH171	I	20	N.A.	4-1BB	1 × 10^6^– 1 × 10^7^	CP	N.A.	N.A.	N.A.	±lenalidomide EGFRt (suicide gene)	
NCT03274219/CRB-402	bluebird bio	bb21217	I	72	murine	4-1BB	150, 300 or 450 × 10^6^	CP/ Flu	69%	75%/ 4.2%	15%/ N.A.	PI3K inhibitor bb007 during ex vivo culture to enrich the drug product (DP) for memory-like T cells	[77]
NCT03288493	Poseida Therapeu-tics, Inc.	P-BCMA-101 (CARTyrin)	I/II	220	human	4-1BB	0.75–15 × 10^6^	CP/ Flu	N.A.	N.A.	N.A.	stem cell memory T cell subset; Rimiducid (safety switch activator) can be administered as indicated	
NCT03338972	Fred Hutchinson Cancer Research Center	FCARH143	I	28	human	4-1BB	50–150 × 10^6^; potential second dose	CP/ Flu	N.A.	N.A.	N.A.	EGFRt (suicide gene); infusion of CD8+ and CD4+ T cells in a 1:1 ratio	
NCT03361748/KarMMa	Celgene	bb2121/ Ide-cel	II	149	murine	4-1BB	150–450 × 10^6^	CP/ Flu	73%	84%/ 5%	18%/ 3%		[6,7]
NCT03430011/EVOLVE	Juno Therapeutics, Inc.	JCARH125 /Orva-cel	I/II		human	4-1BB	lower: 50 or 100 × 10^6^, higher: 300, 450 or 600 × 10^6^	CP/ Flu	91%	N.A./ 2%	N.A./ 4%	1:1 CD4/CD8 ratio preselected prior to transduction and expansion	[78]
NCT03548207/CARTITUDE-1	Janssen Research & Development, LLC	JNJ-68284528/ LCAR-B38M/ Ciltacabtagene autoleucel (Cilta-cel)	Ib/II	126	alpaca	4-1BB	0.75 × 10^6^	CP/ Flu	97,9%	95%/ 4%	21%/ 9%		[72,73]
NCT03602612	National Cancer Institute (NCI)	FHVH33	I	31	human	4-1BB	0.75–12 × 10^6^	CP/ Flu	N.A.	N.A.	N.A.	fully human heavy-chain variable domain	[79]
NCT03758417/CARTIFAN-1	Nanjing Legend Biotech Co.	JNJ-68284528/ LCAR-B38M/ Ciltacabtagene autoleucel (Cilta-cel)	II	60	alpaca	4-1BB	N.A.	N.A.	N.A.	N.A.	N.A.		
NCT04133636/CARTITUDE-2	Janssen Research & Development, LLC	JNJ-68284528/ LCAR-B38M/ Ciltacabtagene autoleucel (Cilta-cel)	III	160	alpaca	4-1BB	0.75 × 10^6^	CP/ Flu	95%	85%/ 10%	20%/ 0%	+Lenalidomide/Daratumumab/ Bortezomib/ Dexamethasone	[80]
NCT04309981	Sara V. Latorre	ARI0002h	I/II	36	humanized	4-1BB	fractionated 3 × 10^6^ + second infusion	CP/ Flu	96%	87%/ 0%	0%/ 0%	fractionated (10%/30%/60% with at least 24 h in between) and multiple infusions; higher CD4/CD8 ratio correlated with more stringent CR	[81,82]
NCT05066646/FUMANBA-1	Nanjing IASO Biotherapeutics Co., Ltd.	CT103A	I//II	132	human	4-1BB	1.0 × 10^6^	CP/ Flu	94.4%	93%/ 2.8%	1.4%/ 0%	enrollment of patients with prior murine BCMA-CAR administrations	[83,84]

^1^ estimated number of patients enrolled or to be enrolled, ^2^ per kg bodyweight, ^3^ CP = cyclophosphamide, Flu = fludarabine, ^4^ completed study results or interim results at the time of publication of this review.

### 2.4. Engineering the BCMA-CAR Construct

Despite all the progress made in construct development, the artificial nature of CARs allows for a very systematic engineering of each compound. Various “adjustment screws” for CARs exist and some of them shall be outlined in the following.

#### 2.4.1. Extracellular Target-Binding Domain: scFV

The standard design for CAR ectodomains is based on monoclonal antibodies. The variable heavy (V_H_) and light (V_L_) chains are linked by a flexible peptide into a single-chain variable fragment. Originally, these antibody domains were of murine origin, like, for example, the BCMA-targeting 11D5-3 scFv in Ide-cel [69,70]. To circumvent potential anti-murine immunogenicity, which must be especially considered if multiple infusions are scheduled, increasing efforts are put into fully human recognition domains. As the causes for CRS and therapy related ICAN are not fully clear to date, attempts to reduce the overall immunogenic potential could lead to increased clinical safety. One example for anti-CD19 CARs in B cell lymphoma treatment is Hu19-CD828Z (NCT02659943), which was assessed in a small clinical trial already and led to lower neurotoxicity compared to the clinically applied axicabtagene ciloleucel [75]. Further reduction of the CAR binding domain size by deletion of the potentially immunogenic linker fragment and consequently reduced immunogenic potential, moreover, led to CAR-designs with a human heavy chain only [79].

Another crucial factor with potential for severe overstimulation of the immune system and thus toxic side effects, is the affinity between scFv and target. For effective CAR signaling and T cell activation, it must not fall below a certain level, on the one hand, but there is evidence, that increased affinity beyond a certain threshold does not lead to increased T cell activation. Instead, this might have disadvantages for longevity. Serial triggering of T cells due to a faster off-rate subsequent to lower affinity is suggested as one possible mechanism for induction of stronger proliferation. Decreasing discrimination between low and high antigen expression on tumor cells with increasing scFv affinity, furthermore, bears a greater risk for systemic toxicity [85,86,87,88]. The other way round, lower affinity scFv depict a possibility to manipulate differential targeting of tumor versus healthy tissue: investigations on a trastuzumab-based CAR-construct in response to HER2^+^ breast cancer showed specific CAR-T cell activation encountering malignant cells without recognition of healthy tissue [87]. Reports from a ROR1-specific CAR on the other hand suggest higher anti-tumor efficacy for scFv with higher affinity towards their target [89]. Possibly, a window of optimal affinity exists, though varying antigen densities might complicate the choice of CAR-construct. This window, however, still must be defined more precisely for different constructs–also those targeting BCMA.

#### 2.4.2. Hinge Region/Spacer Domain and Transmembrane Domain

Despite different specificities, one common structural compound of CARs is the spacer element between the extracellular scFv and transmembrane (TM) region. At least certain constructs require this spacer element for stable CAR-expression and activation. It originally comprised the constant IgG1 hinge-CH2-CH3-Fc-domain. Especially target epitopes close to the cell membrane seem to depend upon a certain degree of flexibility of the CAR which a spacer element provides [90,91,92]. At the same time, this exposed IgG-CH2-Fc domain poses the risk of off-target activation and thereby activation induced cell death (AICD) of CAR-T cells binding IgG Fc-receptors (FcγR) on innate immune cells–which likewise activates those. Reduced CAR-T cell persistence, a lack of anti-tumor control and off-target toxicities are the consequences [92]. Therefore, further adaptations were made: the deletion or modification of distinct regions in the CH2-domain of the spacer for FcγR-binding can effectively reduce this interaction [93,94]. Yet, the exact optimal spacer length, varying from 12 to 299 amino acids for CD19-CARs, for example, depends on the distinct location of the target epitope and most likely the construct [91,95,96]. As an alternative to these IgG-derived spacer/hinge regions native motifs from CD28 or CD8α can be used [97,98]. Regarding, for example, Ide-cel (bb2121), this BCMA-CAR is equipped with the hinge region of CD8α [69].

The TM-region—generally a hydrophobic α-helix spanning the cell membrane anchoring the CAR—of bb2121 is obtained from the natural sequence of CD28 [69]. CD4, CD8α and CD28 have been extensively used to design numerous CARs [99,100,101]. Meanwhile, the TM-domain of inducible T cell costimulator (ICOS) even enhanced T cell activity in one report [35] and thereby underlines, how such fine manipulations of the construct can immensely impact overall CAR-T cell functionality. It becomes evident, that the role of these different TM-regions for T cell activation, in general, is incompletely understood to date and requires further investigation.

#### 2.4.3. Intracellular Signaling Domain

The CAR-part, which probably experienced the greatest engineering already, is the intracellular signaling domain whose modifications define the CAR-generations. As outlined above in more detail the incorporation of either CD28 or 41BB signaling domains can significantly alter T cell activation threshold and longevity. Phosphoproteomic analyses of CD28ζ- versus 41BBζ-CARs pictured the phosphorylation of similar signaling intermediates for both, yet to a larger extent for CD28ζ-constructs [102]. When the same group, Salter et al. [103], performed similar analyses between ROR1-CAR and EBV-specific TCR-activated T cells, they showed fewer phosphorylation of the canonical T cell signaling molecules CD3δ, CD3ε and CD3γ as well as linker for activation of T cells (LAT) for CAR cells. Supraphysiological phosphorylation of CD3ζ and CD28 are suggested to compensate for this lack. These investigations nevertheless describe major differences in signaling cascades between CAR and TCR [103].

Though being able to induce tumor cell killing downstream artificial receptor constructs, the signaling cascade leading there still does not entirely mimic physiological TCR-induced T cell activation. Which consequences do these artificially created signaling differences have for T cell qualities going far beyond immediate activation and anti-tumor control such as T cell differentiation towards terminal effector cells, memory formation and longevity? How is, for example, T cell metabolism altered? Questions like these demand further investigation to get a better understanding of potential harms and to find optimization strategies for CAR-T cell therapy. In this respect, such phosphoproteomic comparative analyses between CAR and TCR at different signaling strengths as well as more detailed metabolic comparisons might be useful for CAR design. Salter et al., for example, concluded an optimized CAR-construct aiming at increased phosphorylation of LAT from their studies by inserting the CD3ε or GRB2 signaling motif. These modifications improved CAR sensitivity, anti-tumor activity and T cell persistence [103]. Other groups are focusing on altering the ITAM domains of CAR constructs for fine-tuned T cell activity by either eliminating them for over-activating CD28ζ or introducing them for less antigen-sensitive 41BBζ CARs [104,105].

Another way of modifying CAR signaling is the addition of cytokines or chemokines: while the current clinical success rates were mostly reached with second-generation BCMA-CARs, first-in-human data are collected with fourth generation constructs with expanded intracellular domains. One example is a BCMA-targeted CAR including expression of IL-7 and CCL19 which was reported superior concerning expansion, differentiation, migration and cytotoxicity while proving safe and efficient in the first patients (NCT03778346) [106].

#### 2.4.4. Tonic Signaling

One further important aspect to be considered in the choice and design of the whole CAR-construct is antigen-independent signaling in the absence of cognate ligand and other exogenous stimuli (e.g., cytokines or allogenic feeder cells) mostly referred to as “tonic signaling”. Consequently, elevated constitutive secretion of cytokines, persistent proliferation for several month without further addition of stimuli, more rapid exhaustion and impaired anti-tumor effects occur [107,108]. At least partly, these effects can be explained by high surface expression levels of CARs caused by strong constitutive promotors and subsequent CAR-clustering at the T cell surface [35]. Higher basal activation and a more differentiated phenotype correlate with higher CAR-surface expression as well as a higher percentage of terminally differentiated effector cells. High surface expression of CARs moreover was enriched in non-responders [109]. Moreover, there is evidence, that the insertion of the CAR-DNA into the endogenous TRAC locus—similarly to orthotopic TCR replacement (OTR)—by homology-directed recombination can prevent tonic signaling. Ameliorated internalization kinetics are suggested to delay effector differentiation and exhaustion then [110], but the procedure might also decrease T cell longevity as other reports showed [111].

The specific construct also influences the strength of tonic signaling: CD28ζ leads to higher, 41BBζ lower levels of tonic signaling. Considering the subsequently more rapid exhaustion and terminal effector differentiation of T cells, this could provide one reason for increased persistence of 41BBζ-CAR transgenic cells compared to CD28-costimulation [112]. Other, earlier reports, however, described antigen-independent proliferation in 41BBζ-CARs as potentially beneficial for longevity in vivo [32]. Tonic signaling per se in T cells is a common phenomenon: TCRs of quiescent cells transiently bind with low affinity to self-peptides in peripheral lymphoid organs causing subthreshold activation by low level phosphorylation of CD3ζ. The relevance of these contacts still remains controversial, but they are expected to impact maintained TCR-sensitivity [113]. This raises the question of whether a defined window of such low, constitutive, or maybe fluctuating T cell activation might in fact be necessary for T cell persistence. Too strong chronic T cell activation including tonic CAR-signaling, however, most likely impairs T cell functionality and must be assessed systematically for different CAR-constructs.

#### 2.4.5. Reducing Toxicities

Overstimulation cannot only affect the transgenic T cells, but their excessive signaling and other immune cells reacting to ACT can also lead to strong cytokine release and is known as one major toxicity of CAR-T cell infusion. Clinically, CRS manifests as fever within hours or days after infusion followed by sinus tachycardia, hypotension, depressed cardiac function, dyspnea and hypoxia. Inflated circulating cytokine levels can entail capillary leak syndrome and consequently pulmonary edema causing lung failure [4,39,70,114,115,116]. Ill-defined and highly heterogenous neurological toxicities (ICAN) can also succeed CAR-therapy and almost exclusively occur in patients who also developed (mostly beforehand) CRS. Correlation with high pretreatment tumor burden and high peak CAR-T cell expansion is moreover described [4,39,114,116,117,118]. Furthermore, several other end-organ toxicities—most of them reversible—and especially hematologic toxicities are reported subsequent to CAR therapy [116].

However, especially the severity of CRS and ICAN demands a better understanding of the pathophysiology and the identification of prediction markers. Further experience with CARs in the clinic will help to improve the management of these side effects [4,119,120]. Nevertheless, reduction or abrogation of toxicities through T cell-engineering strategies should be the ultimate goal. Exemplarily, introducing suicide genes into CAR-T cells depicts one option to eliminate these engineered cells permanently and irreversibly from the circulation by administration of defined drugs targeting these genes in case of severe side effects [121]. Furthermore, several approaches aim at an inducible and reversible switch inhibiting CAR-T cells via administration of drugs such as the IMiD lenalidomide [122] or Doxycycline [123]. Thereby, circumventing the need to destroy CAR-T cells and instead applying more precise control over dose and toxicity due to reversibility might be advantageous.

## 3. The TCR-Based Therapy Approach and Its Potential for Treatment of MM

As CARs are based on our understanding of T cell activation downstream TCRs, ACT cannot only employ artificially constructed receptors, but also native TCR sequences to target a tumor. This strategy, despite being less-well established for clinical application to date, is also promising for MM.

### 3.1. General Aspects of the TCR-Peptide-MHC Interaction

The TCR (see Figure 1A) is the naturally encoded main trigger for clonal expansion of T cells upon specific antigen-binding [124]. Most TCRs are heterodimers composed of disulfide bonded α- and β-chains, both consisting of a constant and a variable domain [125]. The latter, containing the antigen-recognition site, underlies somatic recombination of the variable (V), diversity (D)—only for the β-chain—and joining (J) sequence on the genetic level during its maturation. This leads to a stochastic amount of more than 10^13^ different TCR-clonotypes for each human [126] of which usually only the non-self-reactive survive negative thymic selection [127]. These TCRs recognize with immense specificity their antigen presented on the surface of a tumor or antigen-presenting cell (APC) on a major-histocompatibility complex (MHC) class I or II—the counterpart for either CD8- or CD4-derived TCRs [128]. Compared to the synthetically engineered CARs, the TCR complex lacks an own intracellular signaling domain and only upon association with the six subunits of CD3–CD3εγ, CD3εδ and CD3ζζ—the necessary binding motifs (in total ten ITAMs) for the intracellular signal transduction machinery are provided (see Figure 1A). Following conformational changes of the TCR upon recognition of the cognate peptide presented in the binding cleft of the MHC-complex [129] a variety of different pathways downstream the phosphorylation sites of these ITAMs promotes T cell activation [21,130]. TCRs are distinguished by their enormous sensitivity: It has been shown, that one single or at least very few TCR-pMHC interactions (other studies vary between three and 200 contacts) are sufficient to effectively provoke cytokine secretion and target cell killing by the formation of TCR-clusters triggering serial activation events [131,132,133]. However, usually a much larger number of TCRs is present on the cell surface detecting its respective pMHC-counterpart, reflected by TCR avidity [134]. Generally, T cell clones bearing TCRs of higher functional avidity which is defined as a T cell activation threshold for its effector functions determined in vitro in dependance of a certain epitope density all co-signaling taken into account [134,135], are expected to elicit stronger anti-viral [136,137,138,139] or anti-tumor [140,141,142,143] responses. These cells do not only react more potently to lower doses of antigen in terms of effector functions, but they also seem to lyse their targets faster irrespective of antigen dose [144]. Keeping this higher activation level in mind, one would expect high avidity TCR clones to easily outcompete those of lower avidity on a population level [145,146]. This, however, does not necessarily seem to be the case: For CD4+ T cells it has been shown with respect to affinity, that TCR with a lower level of binding strength to their pMHC-complex and thereby most likely extent of activation, are at least as frequent as TCR clones with higher affinity and contribute to the overall adaptive immune effector function [147]. In addition, there is evidence for activation and expansion capacity of CD8+ cells in response to acute, microbial infection at very low affinity pMHC-TCR interactions [148]. Despite or maybe because of diminished overall activation, T cell clones with lower affinity TCRs persist at higher frequencies after ACT compared to high affinity TCRs in some studies in humans [149,150]. They might maintain a lower and more persistent, rather than strong and pulsatile stimulation resulting in beneficial proliferation patterns [151]. Meanwhile, very high binding strength leads to deteriorated signaling, such as poor mitogen-activated protein kinase (MAPK) phosphorylation [151]. Isolating neoantigen-reactive MHC-class I-restricted TCR from CD8+ T cells from a malignant melanoma patient, we also detected a higher overall frequency of TCR of lower functional avidity in different patient tissues over the course of several years [152,153]. All these findings suggest a “goldilocks” window of optimal T cell activation for in vivo application [154,155]. The strength of T cell stimulation determined by TCR activation threshold might thus imprint the T cell phenotype and thereby functionality, which might be of therapeutic relevance [148,156].

However, additional factors likely play a role. The clearance or persistence of the cognate antigen in settings like either acute or chronic disease—infections as well as tumors—may influence the role of avidity for TCR clones. Moreover, the TCR requires co-stimulation in addition to the antigen-specific signal [157]. It is provided by the integration of a multitude of different stimulatory or inhibitory receptor-ligand interactions, which strengthen or dampen T cell responses. Co-signaling from receptors upregulated in different phases of T cell activation to various extents in dependency of stimulation strength [158] exerts crucial roles in differentiation fate, effector function, proliferation and longevity. The two largest families of co-receptors comprise the immunoglobulin superfamily (IgSF) on the one hand and the tissue necrosis factor receptor (TNFR)-superfamily (TNFRSF) on the other hand [159]. CD28 derives from the first, CD137/41BB from the latter—from those two the previously mentioned most frequently used intracellular signaling domains used in CARs originate. Further understanding of the exact integration of these signals in T cell differentiation, activation, effector response and survival as well as their versatile response to the surrounding tumor microenvironment (TME) and the present ligands will help to make use of co-signaling events in T cell engineering as potential “adjusting screws”.

### 3.2. Potential Targets for TCR-Based Therapy in MM

While antigen recognition by CARs only allows the detection of surface antigens, the choice of target is more diverse and highly personalized in TCR-based therapy. One part of TCR antigens derives from protein-sequences undergoing a multi-step intracellular processing and presentation machinery [160,161]. More recently also non-coding and intronic regions, as well as post-translational modifications, became known as a source for epitopes in the immunopeptidome of a cell [162,163]. Both allow in addition to extracellular targets access to the intracellular peptidome—one major difference and advantage compared to CAR therapy. However, the restriction of peptide-presentation to a highly patient-individual HLA repertoire, renders TCR-based approaches less broadly applicable than CARs. Currently, TCR-based therapies already find use fighting chronic infections, viral [164,165,166] as well as fungal [167], and especially in anti-tumor treatment. For the latter, two options arise in principle: targeting the MHC-presented epitopes of common tumor-associated antigens (TAA)—basically similarly to CARs—or the identification of neoantigens presented individually per patient.

The first option comprises various TAAs: cancer-testis antigens (CTAs), those derived from wildtype proteins but overexpressed in malignant cells as well as lineage-restricted proteins. CTAs distinguish themselves by the restriction to gametogenic tissues with meiotic function as well as pathologically certain tumors [168]. Several CTAs are expressed to different extents depending on the patient and disease stage in MM (see Table 3) and depict potential targets for TCRs: amongst these is the most immunogenic CTA NY-ESO-1 expressed in one third of stage III myeloma patients. To name some more examples, members of the MAGE and GAGE families, LAGE-1 and SSX-2 are also present in malignant plasma cells of which especially MAGEC1/CT7, MAGEA3/6 and LAGE-1 represent promising targets for immunotherapy with a coverage of 85% of MM patients. Furthermore, there is a correlation between the number of different CTAs expressed and the prognosis described [169,170]. However, there is limited information about presentation of defined peptides derived from CTA in MM.

Concerning overexpressed wildtype proteins as TAAs in MM, all surface proteins previously listed as potential candidates for CAR-based therapy, also depict possible targets for TCR-based approaches, as they are generally (over)expressed in MM cells (see Table 1 and Figure 1E).

The limitation of targeting wildtype proteins is their presence in healthy human tissue. On the one hand, on-target, off-tumor toxicities can result if the antigen is expressed—even at very low levels—in healthy human tissue [174]. On the other hand, due to the mechanisms of central tolerance, the natural TCR repertoire is specifically designed to spare self-antigens. Therefore, if at all, only a small amount of TAA-targeting TCR should remain in the T cell pool, which is furthermore expected to have a low affinity towards their pMHC and exert lower anti-tumor function [175,176].

Clinical data, however, show that adoptive TIL therapy, often in combination with checkpoint inhibitor blockade, can induce potent responses in melanoma patients suggesting the presence of tumor reactive TCR-clonotypes [177,178]. Nowadays, it is suggested, that large parts of these therapy-induced or -enforced T cell reactivities could in addition to recognition of TAAs be based on their specificity for neoantigens [179]. These peptides presented on MHC-complexes on the tumor cell surface originate from tumor- and patient-specific, nonsynonymous somatic mutations. Missense mutations, insertions, deletions (indel mutations), frameshift mutations and gene fusions entail these alterations and posttranslational processing as well as antigen presentation moreover influence changes in the peptidome [180,181,182,183,184]. As these novel epitopes are foreign by nature and exclusive for the tumor, they embody optimal target candidates for ACT [185]. Yet, the direct identification of these neoepitopes beyond in silico predictions (e.g., by mass spectrometry) and then eventually the responsive TCRs, remains difficult, so far.

In the past, most studies on the MS-based identification of neoepitopes were conducted in highly immunogenic metastatic melanoma [153,180,186], while to date neoantigens were predicted for several entities, such as for example non-small cell lung, breast, ovarian or gastrointestinal cancer [187,188,189,190]. Especially for more clonal tumor entities with high mutational burden, an increasing number of tumor-recognizing T cells following ICB suggests a major role for the neoantigen-reactive T cell pool in immunotherapy. For these diseases, thus, mutational and neoantigen load correlate as tissue-agnostic predictive biomarker with therapy response rates and survival [179,187,191].

This correlation, however, is not reported to this extent for malignancies with lower mutational burden and higher tumor heterogeneity, such as MM. Is there, though, the potential to exploit neoepitopes for ACT in MM? Whole-exome and RNA sequencing analyses from MM tumor material detected several hundred non-synonymous somatic mutations per patient [192,193,194], of which, depending on the cohort, an average of roughly 20 [194] to 150 [193] could be presented on an HLA-molecule on the tumor cell surface according to prediction algorithms [195,196]. Different myeloma subgroups differ in their overall mutational load, with highest rates for t (14;16) MM. This suggests a different potential for neoantigen-based therapies for these groups [197]. The predicted neoantigen load generally increases with the overall number of genetic alterations as well as in the course of the disease from newly diagnosed to a relapsed situation. Higher neoantigen burden, thus, correlates with decreased survival of MM patients, as could be concluded from the Multiple Myeloma Research Foundation (MMRF) CoMMpass (NCT01454297) study cohort [193,194,197]. Nevertheless, especially among these risk groups of myeloma patients, higher quantity of putative neoepitopes suggests a potentially higher success rate for immunotherapeutic approaches including TCR-based ACT. Indeed, Perumal and colleagues could show for three different relapsed MM patients, that neoantigen-specific CD8+ T cell responses were enhanced by checkpoint therapy [193].

The mutational profile in MM is especially coined by translocations involving chromosome 14, single nucleotide mutations of the immunoglobulin heavy and light chain genes as well as genes involved in the MAPK-pathway, such as NRAS, KRAS (especially in pretreated patients) and BRAF. Several studies, however, reported that the majority of putative neoepitopes are private for individual patients, even if the mutation occurred within the same gene [193,194,198,199]. Aiming towards personalized, precision medicine with TCRs, thus, seems more feasible, than creating “off the shelf” therapeutics for relapsed MM patients. Thereby especially patients with no further immunotherapeutic and even CAR-based therapy options might profit from a selection of TCR for T cell engineering.

### 3.3. Application and Engineering Approaches for TCR-T Cells in MM

Compared to the synthetic CARs, the bottleneck for TCR-based ACT remains receptor identification at present. Tumor-reactive T cells are obtained from tumor specimens or blood samples and can be enriched for antigen-specific activity, exemplarily by sorting for CD137+, CD134+ or PD-1+ cells [200,201,202,203]. Afterwards, these cells are tested in vitro for reactivity against peptides from TAAs or neoantigens by either using barcoded pMHC-peptide multimers [204,205] or co-culture systems with autologous or allogenic peptide-pulsed or minigene-transduced APCs [180,206,207]. Originally, ELISPOT assays measuring IFNγ-secretion, or the upregulation of activation markers were applied. Over the past few years, single cell deep sequencing methods became more and more broadly applicable for the identification of TCR sequences expanding throughout these stimulation assays [150,206,207]. Nevertheless, TCR identification remains expensive and time-consuming—at least several weeks from initial mutation calling to TCR isolation are reported in the fastest pipelines currently available [204,208]. Facing the instability of tumor mutations and consequently immunopeptidome, acceleration of these processes is substantial for a broader application.

While CAR-based approaches are already tested in various clinical trials for the treatment of MM, the way for TCR-based T cell products into the clinic for MM therapy as well as for other entities remains more troublesome at present. Some TAAs can be targeted by “off the shelf” TCR-constructs which can be introduced genetically into autologous patient T cells and then reinfused. Yet, these products remain HLA-restricted and large patient cohorts usually needed for drug development are often not feasible the more personalized these therapies become.

For MM, T cells modified to express receptors against epitopes derived from the CTAs NY-ESO-1 or LAGE-1 (NCT01892293, NCT01352286 [209] and NCT03399448 [210]) are clinically evaluated in a small set of patients already. Here, they show responses in high-risk relapsed or refractory HLA-matched MM patients after myeloablation and ASCT (see Table 4). For the NY-ESO-1/LAGE-1 (the target epitope represents a shared peptide of both highly homologous CTAs) specific peptide enhanced affinity receptor (SPEAR) transgenic T cells [211] an ORR of 80% with a median PFS of 13.5 months could be achieved. After one year, 52% of patients remained disease progression-free. T cells were homing to the bone marrow, the tumor site, and showed proliferative capacity and longevity (quantifiable for 100 days in 23 of 25 patients; for two patients even for 5 years). At the same time, the occurrence of severe side effects such as CRS or ICAN remained low as, so far, experienced for TCR-based approaches in contrast to CARs [209].

These SPEAR-TCR were altered in their amino acid sequence before in silico reconstitution to increase affinity [212,213] since the major predictive factor in TCR-based ACT for effective anti-tumor control still is high TCR affinity [213,214]. This affinity maturation reaches an acceleration of the human T cell response towards their cognate antigen yet deteriorates the ability to recognize the target at low densities [215]. As outlined above the suggested “goldilocks” activation strength and decreased T cell effector functions at too high TCR affinity [154,155], suggest an upper limit for this modification, which, however, still has to be defined. No further precise criteria for the “best” TCR for ACT are established to date—if such a best receptor even exists, rather than different qualities necessary at different disease stages, antigen densities or under different TME conditions. The heterogeneity in functional avidity of different TCRs in the anti-tumor T cell pool from melanoma patients as mentioned above suggests, that a deeper understanding of favorable TCR qualities is necessary across different entities [150,152]. Understanding, which TCR clonotypes have the greatest impact on immediate tumor killing, but also long-term tumor control and memory formation, will be essential to choose the optimal combination of TCR for ACT improving patient survival.

Further development of NY-ESO-1-TCR-engineered T cells in the group around Stadtmauer et al., added not only the CRISPR-Cas9 based deletion of the endogenous TCR-α- and β-chain to the lentiviral manufacturing procedure of these cells, but also included further engineering: the knockout (KO) of the immune checkpoint molecule Programmed cell death protein-1 (PD-1). PD-1 deficient T cells were previously shown to exert better anti-tumor function of CAR T-cells in xenograft mouse models [216,217], while the combination of NY-ESO-1 specific TCR-T cells with anti-PD-1 blockade augmented efficacy [218]. Stadtmauer et al. thus performed the first-in-human pilot study for CRISPR-Cas9 engineered TCR-T cells for two refractory MM patients and one sarcoma patient, demonstrating not only feasibility but also the safety of this engineering approach [210]. The KO of PD-1, however, demands a more detailed understanding of PD-1 signaling and the potential negative influences of T cell differentiation. The contraction of putative PD-1 KO-T cells (identified by sequenced editing events in the gene) from 25% to 5% of all T cells in four months within the study, rather suggests a deficit in longevity than the advantage expected from previous publications [219]. PD-1, moreover, was identified as a haploinsufficient tumor-suppressor for T cells suggesting a certain degree of oncogenic potential of PD-1 KO [220], why such engineering strategies must be applied with caution.

Apart from NY-ESO, other TAAs currently targeted in preclinical investigations are the B cell-specific transcription factors BOB1, FCRL5 and VPREB3 showing anti-tumor efficacy against several B cell malignancies, amongst them MM [221,222]. MAGE-A1 specific TCR are moreover evaluated clinically for a cohort of refractory MM patients in a German study (DRKS00020221). Additional clinical trials evaluating in-patient responses are, however, still necessary to further assess therapeutic potential here.

While these TAAs, if overexpressed in their wildtype form in the tumor, allow “off the shelf” TCR-constructs with restriction only to the matching HLA—this already highly limits the feasibility of large-patient cohorts—targeting neoantigens is highly personalized. The extremely patient-specific nature of most epitopes recognized on top of the TCR’s HLA-restriction claim for single-patient protocols promising high specificity and minimized toxicity. Despite the preclinical success and progressive identification of various neoantigen-specific TCR with anti-tumor effector functions [150,152,153,207], no results from clinical trials are available, so far. Currently, one single-arm first-in-human phase Ia/Ib trial is open and recruiting patients for investigating the efficacy of a single dose of neoantigen-specific TCRs with and without additional anti-PD1 treatment in locally advanced or metastasized solid tumors (NCT03970382).

It will depend on results from studies like this, whether such a highly personalized and thereby surely laborious therapeutic approach, becomes more feasible and will be expanded to more entities such as relapsed patients with MM. The hypothesis is that the potentially broader range of extra- and intracellular immunologically relevant epitopes targeted via private TCRs compared to CARs targeting one single surface antigen, might be a promising therapeutic option for patients refractory to “off the shelf” products due to tumor escape mechanisms such as antigen loss.

## 4. Therapy Resistance: Obstacles for CAR as Well as TCR T Cell-Based Therapy Approaches in MM and Engineering Strategies

Despite the clear potential CAR- as well as TCR-based therapeutic approaches promise for MM further optimization is required on various levels to efficiently eradicate MM cells. The players in this game are, on the one hand, the tumor cells with a certain antigen or peptide-MHC complex presented on their surface as well as a tumor-specific, highly immune-suppressive microenvironment. On the other hand, we redirect T cells towards this particular antigen by genetically engineering them to express a specific receptor to target and kill the tumor cells. We are aware that this is, of course, a vast simplification of the complex cellular interplay between the innate and adaptive immune system as well as the surrounding tissue. Even though, we first need a deeper understanding of how to optimize T cell engineering strategies on these “simple” three levels: the receptor itself, the isolated, transduced T cell population and the tumor with its surface expression and a defined microenvironment (see also Figure 1). Several mechanisms on each of these levels can lead to therapy resistance. We already elaborated on strategies for fine-tune engineering on the level of CAR- and TCR-constructs themselves and thereby improving qualities such as target recognition, T cell activation, tumor cell killing, toxicity profile and longevity. In the following, we want to shed more light onto the other two, equally important levels.

### 4.1. Counteracting Unfavorable T Cell Intrinsic Qualities

The infusion product is made from living cells with different in vivo fates prior to transduction—often various pretreatment lines fighting the malignancy. Each cell product, thus, is distinguished by its highly individual properties. Beneficial subsets and markers have been mostly described for CAR T cells but can be, at least in part, transferred into the TCR-ACT setting.

In vitro transduction and expansion processes as well as antigen-specific T cell stimulation in the patient afterwards lead to differentiation towards a terminal effector state with reduced multipotency and overall functionality [223,224]. Therefore, the enrichment for less differentiated T cells—naïve (TN), stem cell memory (TSCM) or central memory (TCM) phenotypes—leads to improved T cell persistence in the patient [225,226,227]. Furthermore, retarding terminal differentiation of T cells by the addition of PI3Kδ-inhibitors during in vitro culture restored functional capacity [228]. Autologous T cells, however, might already be dysfunctional consecutively to disease progression—one hallmark of cancer—or exposure to chemotherapeutic agents. To circumvent cellular malfunction, one further possibility—also critically accelerating the time between therapy indication and administration—might be allogeneic T cell therapeutics as discussed elsewhere in more detail [229]. Tightly balancing the potential of increased T cell fitness and “off the shelf” availability on the one hand, with graft-versus-host disease, on the other hand, these approaches demand for extremely precise matching of the highly polymorphic HLA molecules between donor and recipient or effective T cell engineering including for example HLA-modification or inclusion of off-switches. So far, these approaches are especially investigated for CAR-T cells [229,230]. Nevertheless, the stimulation, these T cells experience during manufacturing processes and in the patient, leads to terminal differentiation and senescence as well as counterregulatory processes often termed exhaustion. In both cases, diminished effector capacity results, which leads to diminished anti-tumor response. Exhaustion has been suggested as a major reason for T cell dysfunction in the setting of chronic T cell stimulation due to antigen persistence instead of clearance like in acute infections—chronic viral infections, as well as tumors, provide this condition. In response to continuous TCR signaling T cells upregulate inhibitory receptors such as PD-1, CTLA-4, Tim3, Lag3 or TIGIT and loose effector function [231,232]. Since high expression of inhibitory receptors like PD-1 is found in TILs, they are expected to be exhausted already—the reversibility of these processes is still controversially discussed [233,234,235]. Consequently, the use of TCR-transgenic T cells instead of expanded TILs for ACT is already a strategy to circumvent impaired functionality. However, of course, adoptively transferred T cells also face the same persistent antigen stimuli as endogenous tumor-specific T cells and are prone to become dysfunctional in the course of this counter regulation as well. Developing strategies against exhaustion, such as a combination of TCR- (e.g., clinically tested in NCT03970382) and CAR-T cells with checkpoint inhibitors [236] or genetic modification such as the KO of PD-1 [210] may represent important steps towards long term anti-tumor control subsequent to ACT.

In the opposed case—extremely low or absent antigen-specific stimulation—the number of unstimulated T cells rapidly declines after infusion due to the lack of expansion triggers. The combination of CAR-engineered T cells with RNA vaccines for body-wide presentation of the target epitope in all lymphoid compartments, thereby may bypass insufficient engraftment by providing continuous stimuli to transgenic T cells [237,238].

T lymphocytes, however, do not only differentiate subsequently to stimulation, but already commit to either CD4 (MHC-class II restricted) or CD8 (MHC-class I restricted) lineages in the thymus [239]. CD4+ T-cells are considered “helper cells” interacting with professional APCs and secreting cytokines to provoke immune responses and attract other immune cells. CD8+ T cells are also called “killer” or cytotoxic T cells. While direct tumor cell killing is mainly attributed to the latter, the former hold important accessory functions for anti-tumor immunity [240,241,242]. Antitumor reactivity was indeed enhanced by infusing precisely defined compositions of CD4 and CD8 T cells for transgenic cell products [243] and these combinations are currently evaluated further in the clinic for CD19-CAR therapies (NCT01865617 and NCT01865617) [114,118]. Similar combinations also have to be tested for TCR-based therapeutics, limited by the necessity for parallel discovery of target peptides and matching TCR for MHC-class I and MHC-class II.

### 4.2. Tumor Resistance Related to Antigen Expression

Stable expression of the antigen of interest on the cell surface—either as a whole antigen or as a peptide on an MHC-complex—and its coverage of the tumor entirety before therapy initiation determine therapeutic response. For BCMA [93] and even more pronouncedly for neoepitopes, it is highly unlikely, that all tumor cell clones express the target of interest to the same extent at diagnosis and the start of therapeutic intervention. Generally, a large degree of intratumoral heterogeneity is expected before the start of treatment already [199] and therapeutics targeting antigens with incomplete coverage, thus, specifically select for target-negative and thereby therapy-resistant tumor cell clones. The development of reliable strategies and methods for intratumor-heterogeneity assessment at therapy baseline, therefore, is essential for successful long-term tumor control [199].

Regarding stability and coverage of the BCMA-surface expression in MM, its level can also be highly variable. Heterozygous BCMA loss before treatment initiation, for example, is reported for some MM cases and can also occur later throughout therapy-induced immunoediting as the clonal selection pressure largely comprises stability of surface expression [93,244]. Each slightest alteration in the expression of the target in response to therapy potentially leads to refractory tumor residues which are resistant to the adoptively transferred cells [245]. Moreover, the BCMA surface-level seems largely heterogeneous for different myeloma cell lines as well as patients [246,247] and can fluctuate within one cell line depending on in vitro culture conditions [unpublished own data]. This suggests cell-intrinsic mechanisms for up- and downregulation in response to environmental stimuli. Cleavage mechanisms of the ubiquitous multi-subunit γ-secretase complex are, exemplarily, known to reduce target density on the MM cell surface resulting in a soluble BCMA (sBCMA) form [248]. Consequently, inhibiting BCMA cleavage augmented antigen surface expression and CAR-T cell mediated antitumor activity while proving safe and tolerable in first clinical trials [249,250]. Especially for CARs, which are known to be much less sensitive than TCR [103], the sensitivity of the receptor-construct must therefore be considered for antigen choice: Low or decreasing levels of BCMA-expression beneath the detection limit can already lead to failure of CAR therapy.

For TCR-based approaches, tumor cells become unrecognizable for the T cell compartment, for example, by downregulation, loss and/or mutation of the antigen processing and presentation machinery, like the beta-2-microglobulin domain of the MHC-I complex [251,252]. In this manner, antigen-negative subclones are selected by the therapies applied according to Darwinian principles. Likewise, other immune therapies are also shown to alter the surface expression landscape of tumor cells: ICB, for example, leads to loss of certain neoantigens probably due to loss of tumor subclones. Consequently, this can result in acquired resistance to ICB and influences therapeutic decisions for the administration of ACT [253].

Taking these tumor escape mechanisms into consideration, a panel of several targets for ACT could be a potential solution to reduce the risk of tumor escape. For CAR-based approaches, on the one hand, several strategies can be followed: to avoid the risk of therapy resistance due to structural BCMA-alterations, biepitopic CAR-constructs are currently developed and tested [72,254,255]. Counteracting the complete loss of BCMA during CAR-treatment, cocktails or sequential administration of CARs with several surface target antigens (see Table 1) are furthermore evaluated in other entities already [256].

For TCR-based strategies, on the other hand, the enormous range of possible and detectable neoepitopes, their patient-specificity and intratumor heterogeneity complicate target selection. The prioritization of candidate neoantigens usually is based on a diversity of aspects such as their detection by mass spectrometry or predicted binding affinity to the MHC molecule, as well as the number of reads including the alteration in NGS data. Neoepitopes binding to several HLA-molecules might be moreover beneficial for stable presentation when facing mutations in the antigen-presentation machinery. The most important features of neoepitopes, thus, are the likelihood of their presentation on the tumor cell surface and the expected T cell response they elicit [257,258,259,260]. In addition, qualitative characteristics, such as homologies to infection-derived peptides were shown to correlate with longer survival in pancreatic cancer patients [259]. This is in line with findings of a correlation between better therapy response to PD-1 checkpoint blockade and higher dissimilarity of predicted neoepitopes to the self-proteome of these patients [261].

Despite all progress made with in silico predictions of neoepitopes, it remains ill-defined, which particular epitopes lead to strong immune responses, which remain neglected by the immune system and which can and should be therapeutically exploited [262]. Various studies underline the complexity of the immunogenic potential of different tumor epitopes and demonstrate, that algorithms still lag behind: ex vivo testing of previously established T cell responses against neoantigens shows potential for either anti-tumor efficacy or immune-inhibitory function [263]. Furthermore, even if in vitro anti-tumor response can be measured in response to a neoantigen, this does not necessarily mean antigen recognition and T cell activation in vivo, where numerous other factors such as the TME have to be considered.

### 4.3. MM-Defined Suppression of T Cell Action

The T cells’ opponent, the tumor entity, of course, immensely influences their fates. Especially for solid tumors, from which TILs can be directly isolated, it is known, that the presence of potentially tumor-reactive T cells is not necessarily accompanied by tumor rejection. This dysfunction of tumor reactive T cells after their infiltration into the highly immunosuppressive TME has been described by Hellström and colleagues [264,265]. In a similar fashion, the immunosuppressive bone marrow (BM) niche in MM can silence cytotoxic activity of reactive T cell clones [266,267]. While the composition of immune cells in this BM niche has been reviewed elsewhere in more detail [268], some immunosuppressive factors with special relevance for MM shall be highlighted here.

On the one hand, crosstalk between T cells and other tumor-promoting immune cells in the tumor niche can lead to dysfunction of T cells. Various interactions contribute here, but one should be briefly elaborated on in the context of ACT: regulatory T cells (Tregs) defined as CD4 + CD25 + FOXP3 + play an important role for immune tolerance to self-antigens. However, by competitive IL2-consumption, immunosuppressive cytokine secretion (IL-10, TGF-β) or suppression of APCs via CTLA-4 they can also reduce anti-tumor immunity [269]. The balance between anti-tumor activity by cytotoxic CD8+ T cells and immunosuppression by CD4+ Treg cells is considered a key driver of progression in myeloma from the preliminary monoclonal gammopathy of undetermined significance (MGUS) to MM by many reports [270]. In some investigations, either elevated absolute Treg numbers or an imbalance in the ratio of Treg to TH17 or CD4 effector cells are correlated with myeloma progression and impaired clinical outcome [271,272,273]. Others, however, report decreased numbers or dysfunctionality of Treg cells in MM patients as well as alterations in the course of treatment (e.g., with thalidomide). This also suggests one possible reason for inter- and intra-cohort differences in the evaluation of the Treg-role [274,275]. Depletion of Treg cells is suggested to, nevertheless, play a role in MM treatment and represents one possible mechanism of action of anti-CD38 antibodies such as Daratumumab and Isatuximab [276,277].

Thinking about adoptive T cell product infusion, the role of Tregs requires further understanding. Most investigations in this respect have been rolled out with CD19-targeting CARs in B-cell malignancies, so far, but will also be necessary for optimizing administration of CAR- and TCR-based therapeutics in MM. In the first place, the presence of prior established tumor specific Treg populations might impair the anti-tumor response of infused transgenic T cells [278]. Due to the lack of natural Treg cells in xenograft mouse models, these effects remain hard to follow-up [278]. The improvement of ACT approaches by prior lymphodepletion regimens, however, might at least partly be explained by the depletion of these immunosuppressive, regulatory T cell populations [279]. Moreover, the tumor specific activation of CAR- or TCR-T cells itself—including high IL-2 secretion—potentially induces the formation of Treg populations [280,281]. Several strategies are employed, to prevent this. One of them is the disruption of the IL-2 axis by engineering CAR- (and in the future possibly also TCR-) T cells to express the IL-7-receptor [282] or a IL-7R/IL-2β hybrid receptor [283]. This should selectively promote the growth of effector CAR-T cell populations via IL-7. Engineering the co-stimulatory CAR domains—and along this line the CD28-induced LCK phosphorylation and thereby IL-2 secretion—might also influence Treg induction; results though stay contradictory, so far [283,284,285]. Besides, CAR-T cells themselves have been shown to acquire a regulatory-like phenotype in the presence of TGFβ, which depicts one further requirement for engineering tumor-redirected T cells [286].

Apart from MM-specific Treg cells, myeloid derived suppressor cells (MDSCs), B regulatory cells or MM cells themselves can express and secrete immunosuppressive factors impairing T cell functionality [268,287]. Interaction of surface co-signaling receptors especially got into the focus in the past few years. By blocking inhibitory receptors (e.g., PD-1, CTLA-4) or their ligands (e.g., PD-L1) T cell inhibition via these interactions shall be reverted and T cell functionality reinvigorated [288,289]. Despite initially promising in vitro results [290], clinical outcomes of ICB as a single agent in MM remained unsatisfactory, so far [291], while combination with IMiDs even led to unfavorable risk-profiles [292]. This lacking benefit might at least partially correlate with the phenotype of T cells in the BM niche of MM patients. Compared to the clearly “exhausted” phenotype of T cells in the TME of other entities, like melanoma [234], T cells in MM express rather low levels of inhibitory receptors (such as PD-1, CTLA-4, Lag3 or Tim3). Instead, these cells were CD28−, KLRG1+ and CD57+ (see Figure 1F)—characteristics of a late-differentiated, senescent status—and showed impaired proliferation. However, this state was accompanied by normal-for-age telomere-lengths suggesting potential reversibility of this proliferative impairment [293,294].

Hypo-responsiveness of T cells in MM, thus, might require therapeutic strategies other than targeting inhibitory receptor axes alone. Immunomodulatory drugs (IMiDs) are established in standard procedures in MM treatment already and enhance T cell responses by boosting proliferation, increasing IL-2 and IFN-γ secretion (T_H_1 cytokines), downregulating immunosuppressive cytokine secretion and inhibiting Treg formation [295]. It suggests that equally to the advantage on endogenously present tumor-reactive T cells, combinations of IMiDs and CAR-/TCR-based therapies promise improved clinical outcome [296]. Several other combinatorial approaches with antibodies (e.g., anti-CD38 [297]) or armored CAR-transgenic T cells (TRUCs) resisting immunosuppressive factors [298,299,300] are exploited already. They exemplarily demonstrate the immense number of “adjusting screws” and engineering approaches executed in order to optimize T cell-based therapeutic strategies responding to the TME in MM.

## 5. Conclusions

Many important steps have been made in immunotherapeutic treatment of MM patients in summary. Amongst them are, on the one hand, FDA- and EMA-approved BCMA-CAR-constructs for T cell-based ACT. However, to make CAR-based treatment in MM more broadly applicable many more hurdles as discussed in this review need to be overcome: it will be necessary to aim at a better understanding of CAR-signaling as well as increased knowledge of the effect these artificial receptors have on T cell reactivity and longevity. Thus, we need to assess the opportunities, but also difficulties and downsides of certain modifications in these constructs and therapy regimens. On the other hand, we may have to remember that physiological T cell-activation happens downstream a TCR and that the immense sensitivity and specificity, as well as the broad target repertoire, promise personalized approaches with reduced toxicities. Yet, we still lack the ability to identify suitable HLA-presented epitopes and reactive TCR on a large scale and require understanding, which qualities of TCR are most beneficial for which target and entity. For both, CAR as well as TCR, there are many approaches assessed already to face resistance mechanisms. Despite conceptual attractiveness, they still must prove clinical superiority, more durable response rates and prolonged survival in MM patients.

## Figures and Tables

**Figure 1 cells-11-00410-f001:**
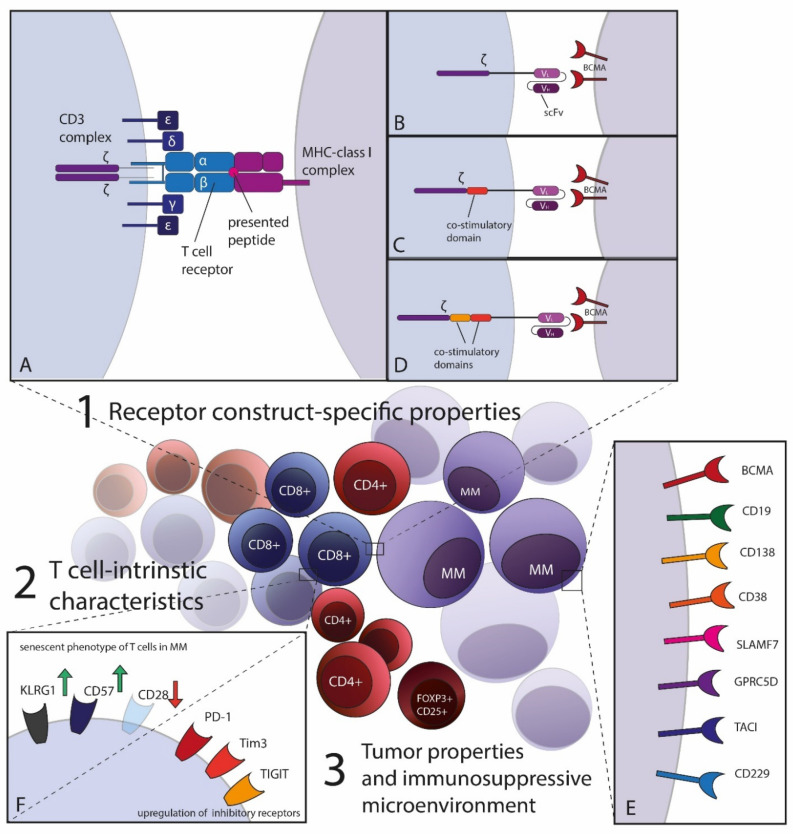
Engineering CAR- or TCR-transgenic T cells in MM. Transgenic CD8+ (blue) and CD4+ (red) T cells are shown encountering MM cells (violet). As an exemplary immunosuppressive element for ACT in MM a FOXP3^+^CD25^+^ Treg cell (dark red) is present amongst the cells. The levels on which T cell engineering can take place (1–3) are indicated. The genetically transferred constructs are depicted schematically in proximity to their target structures—TCR associated with the chains of the CD3-complex (**A**) or CAR (1st (**B**), 2nd (**C**) and 3rd (**D**) generation)—as well as different potential surface targets on MM cells (**E**). The typical surface expression of senescence markers in T cells for MM as well as the upregulation of inhibitory markers is also depicted as a T cell-intrinsic characteristic (**F**).

**Table 1 cells-11-00410-t001:** Surface antigens in MM: potential targets for CAR-based ACT.

Target	Other Names	Physiological Single Cell-Type Enrichment (Proteinatlas.Org)	Identified Ligands (Uniprot)	Involvement in Biological Process (Uniprot)	Car-Based Clinical Trials in Mm (Selection from Clinicaltrials.Gov)
CD19		(naïve and memory) B cells		co-receptor for B cell receptor, B cell activation, proliferation, differentiation and antibody-production	NCT04194931, NCT03706547, NCT04603872, NCT02794246
CD38		ciliated cells, erythroid cells, granulocytes, Kupffer cells, T cells, NK cells	NAD, NADP	production of second messengers cyclic ADP-ribose and nicotinate-adenine dinucleotide phosphate, cADPr hydrolase activity	NCT03464916, NCT03767751, NCT03778346
CD138	Syndecan 1, SDC1	hepatocytes, urothelial cells, cholangiocytes, memory B cells		linking of cytoskeleton and interstitial matrix, regulation of exosome biogenesis	NCT03672318, NCT03778346
BCMA (B cell maturation antigen)	TNF receptor superfamily member 17, TNFRSF13a, CD269	melanocytes, erythroid cells, (naïve and memory) B cells, plasmacytoid DCs)	TNFSF13B/BLyS/ BAFF and TNFSF13/APRIL	B cell survival, regulation of humoral immunity, activation of NF-kappa-B and JNK	see Table 2
Integrin β7		B cells, granulocytes, T cells	Magnesium, Metal-binding	cell adhesion, lymphocyte migration and homing to gut tissue	NCT03778346
SLAMF7/SLAM-family member 7	CS1, CD319	Monocytes		immune cell activation, connection of innate and adaptive immunity	NCT04499339, NCT03958656, NCT03778346
GPRC5D (G-protein-coupled Receptor Class C Group 5 Member D)		early spermatids, melanocytes, late spermatids, B-cells		not yet determined in detail	NCT04555551, NCT05016778
Immunoglobulin light chain		B cells			NCT00881920
CD229	Lymphocyte antigen 9 (LY 9)	melanocytes, B cells, T cells, erythroid cells, plasmacytoid DCs		member of the SLAM-family, activation and differentiation of a variety of immune cells	
TACI (Transmembrane activator and CAML interactor)	TNF receptor superfamily member 13B		TNFSF13/APRIL and TNFSF13B/ TALL1/BAFF/BLYS	stimulation of B and T cell function, Calcineurin-dependent NFAT-activation, NFkB and AP-1	

**Table 3 cells-11-00410-t003:** Frequently expressed cancer-testis antigens in MM patient cohorts.

Gene	Expression in BM-Samples from MM Patients Across Different Studies [in %]				
	Andrade et al., 2008 [169] (n = 39)	Van Duin et al., 2011 [171]		Atanackovic et al., 2007 [172] (n = 55)	Condomines et al., 2007 [173] (n = 64)
		newly diagnosed (n = 320)	relapsed (n = 264)		
BAGE1	32			14.5	
CTAGE5		95.6	48.5		
CTNNA2		60.6	26.5		
FAMI133A		86.3	79.2		
GAGE (family)	36				17
GAGE8		15	61.4		
GAGEA		16.6	71.2		
JARID1B		82.5	33.7		
LAGE-1	49				
MAGE A3/6	46	37.8/45	47.3/49.2	54.5	33/31
MAGE A9		10.9	5.7		
MAGE B1		5.3	3.8		0.9
MAGEA1	31	21.9	42		3.7
MAGEA12	20.5	15.3	33.7		25
MAGEA2	41	9.4	8.3		2.0
MAGEA4		3.1	5.7		0.2
MAGEB2		47.2	27.7		
MAGEB4		5.3	1.1		
MAGEC1/CT7	77	71.3	60.6		66
MAGEC2		29.1	9.5	56.4	13
NY-ESO-1	36			7.3	
PAGE2		5.9	2.3		
PBK		94.1	86.4		
PRAME	23	31.9	37.9		
SPA17		38.1	9.1		
SPAG9		100	99.6		
SPANXC		5	3		0.1
SSX1	28	30.3	29.5	34.5	20
SSX2		6.6	6.4	16.4	0.6
SSX3		2.5	5.7		0.4
SSX5				20.0	
TEX14		7.2	3		
TSPY1		10.6	13.6		
ZNF165		83.1	13.6		

**Table 4 cells-11-00410-t004:** Clinical trials for TCR-based therapy in MM (information from clinicaltrials.gov).

Trial Number/Name	Sponsor	TCR-Specificity	HLA-Restriction	Phase	n	Diagnosis	Dose ^1^	Conditioning Therapy ^2^	Further Modifications	Reference
NCT01892293	Adapt-immune	NY-ESO-1c259	HLA-A*0201	I/IIa	6	relapsed or progressive MM	1 × ≥ 0.1–1 × 10^10^; in case of progression: second dose of up to 5 × 10^10^			
NCT01352286	Glaxo SmithKline	NY-ESO-1c259 (high affinity)	HLA-A*0201	I/IIa	25	relapsed or refractory MM (at least one prior therapy line)	>0.1–1 × 10^10^		affinity maturated TCR/ specific peptide enhanced affinity receptor (SPEAR) T cells	[209]
NCT03399448	University of Pennsyl-vania	NY-ESO-1c259 (high affinity)	HLA-A*0201	I/IIa	3	refractory metastatic sarcoma, relapsed or refractory MM (at least three prior therapy regimen)	1 × 10^8^	CP, FLU	electroporated with CRISPR guide RNA to disrupt expression of endogenous TCRα, TCRβ and PD-1 (NYCE T Cells)	[210]
NCT02457650	Shenzhen Second People’s Hospital	NY-ESO-1	HLA-A*0201	I	36	various entities, amongst them MM	N.A.	CP, FLU		

^1^ per kg bodymass, ^2^ CP = cyclophosphamide, FLU = fludarabine.

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
