# Peer review of "Adoptive Cellular Therapy for Multiple Myeloma Using CAR- and TCR-Transgenic T Cells: Response and Resistance"

_cells, 2022, doi:10.3390/cells11030410_

Round 1
Reviewer 1 Report
This is a well written summary of CAR T cells and TCR transgenic T cells in myeloma. It is comprehensive and is relatively easy to follow.
The main criticism is the length of article spent on TCR transgenic T cells which are still only in very early phase studies and only been used in very limited numbers of patients.
I would prefer to see more details on CAR-T cells. No mention is made regarding the CARTITUDE trials which have progressed to phase 3 or alternate CAR-T cells such as allogeneic CAR-T cells.
I think the article would read better if the explanation on TCR transgenic T cells was summarised more succinctly and discussion on CAR-T cells was expanded. This is particularly important given that CAR-T cells are likely to be more widely available than TCR transgenic T cells.
Author Response
We thank the reviewer for his/her feedback and agree on the opinion that CAR-T cells are currently way more broadly applied in the clinic than TCR-T cells for the treatment of multiple myeloma. As requested, we therefore added more detailed information about the clinical status of CAR-trials (especially an update on the CARTITUDE studies, but also updated CAR-related data from the 63rd ASH Meeting in December 2021). Moreover, we shortened the part spent on general information about TCR-T cells as advised and hope to make the article more concise therewith. However, we would not leave out further information on TCR-based adoptive cellular approaches due to the future potential we see in these engineering approach for different tumor entities, amongst them multiple myeloma. One aim of our review is the comparison of the basic concepts of CAR- and TCR-based therapeutic strategies – knowing, that the first already is tested more extensively in the clinic than the latter to date.
Reviewer 2 Report
This is a very well written, kind of encyclopedic work encompassing the entire spectrum of T-cell therapy as applied to the treatment of multiple myeloma. The text is well readable and even the difficult-to-understand parts are clearly described for the interested reader. CAR-T versus tumor-reactive TCRs of the patients present the two main themes throughout the manuscript. Basic mechanisms are presented with the description of the major differences in signaling cascades between CART and TCR transduction, as they apply to the field of myeloma.
Engineering aspect of the differnt constructs are detailed and provide effective "crutches" for readers who are newcomers to the field. Additionally, the newly emergent field of adoptive T-cell transfer, as related to the tumor peptiome, are detailed. Finally, resistance mechanisms to T-cell therapy are introduced with all available detail.
I have only two minor comments:
- adoptive cell therapy already in clinical use for the treatment of refractory viral (and fungal) infections may need to be mentioned.
- the very receny result updates presented at ASH 2021 need to be incorporated - especially to the tables describing results of the different approaches
One typo: line 254 transtuzumab should be written without an n
Author Response
We thank the reviewer for his/her feedback and adapted the manuscript according to the suggestions made. We agree that the recent updates on CAR-based MM therapy published during the 63rd ASH Meeting in December 2021 are mandatory to mention in our review and included the results in text and tables. Furthermore, we now also comment on the application of TCR-based adoptive cell transfer besides cancer treatment in anti-viral and anti-fungal therapy.